# Compositional Scene Modeling with An Object-Centric Diffusion Transformer

## Abstract

Early object-centric learning methods adopt simple pixel mixture decoders to reconstruct images, which struggle with complex synthetic and real-world datasets. Recent object-centric learning methods focus on decoding object representations with complex decoders, such as autoregressive Transformers or diffusion models, to solve this problem. However, these methods feed all object representations together into the decoder to directly reconstruct the latent representation of the entire scene. Contrary to human intuition, this approach ultimately leads to weak interpretability. This paper combines the recent powerful diffusion model and composition module to propose a novel object-centric learning method called Compositional Scene Modeling with an Object-centric Diffusion Transformer (CODiT). By adopting a proposed compositional denoising decoder that can generate the mask of single objects and construct images compositionally, CODiT has stronger interpretability while still retaining the ability to handle complex scenes. We also illustrate the Classifier-Free Guidance explanation of CODiT. Experiments show how compositional structure helps control the generation process, allowing the model to generate images via single object representations and edit objects. In addition, we present CODiT performs strongly in various tasks including segmentation and reconstruction on both complex synthetic datasets and real-world datasets compared with similar methods.

## 1 Introduction

Visual scenes are often composed of multiple visual concepts. Because of combinatorial explosion, even a few types of objects can generate infinite visual scenes with rich diversity. Therefore, learning individual representation for the whole scene is unsuitable for scene understanding tasks (Santoro et al., 2017). Since all information is entangled in this complex representation, extracting the representations of the objects in the scene that are more meaningful is difficult. Humans can learn and understand visual scenes effectively. One key ingredient of this ability is to perceive the world in a ***compositional*** way (Lake et al., 2017), where humans decompose visual scenes into multiple regions and extract the corresponding representations. In the process of painting or scene modeling, humans also first construct individual objects based on the corresponding memories, and then further complete the entire complex scene compositionally. Object-centric learning (OCL) aims to enable machines to learn and utilize object representations compositionally like humans to deal with the scene diversity caused by the combination of visual concepts (Yuan et al., 2023). In this way, machines can handle complex scenarios more easily, and the extracted representations can be more consistent with human intuition (Yi et al., 2018; Mao et al., 2019).

Existing OCL methods can be divided into two categories, 'post-decoder' compositional (i.e., constructing the scene by composing the individual objects decoded from individual object representations) methods as well as 'pre-decoder' compositional (i.e., constructing the scene by concatenating all objects' representations and putting it into a decoder) methods, as illustrated in Figure 1. The post-decoder compositional methods (Greff et al., 2017; Eslami et al., 2016; Yuan et al., 2019b; Locatello et al., 2020; Engelcke et al., 2021; 2019) often employ a pixel mixture decoder. They generate the object masks and appearances separately and adopt the objects' masks as mixture weights to combine the objects' appearances for image construction. Such models have stronger compositional properties and interpretability but have been observed (Singh et al., 2022a; Seitzer et al., 2022) to struggle in complex scenes. In contrast, the pre-decoder compositional methods have fo-

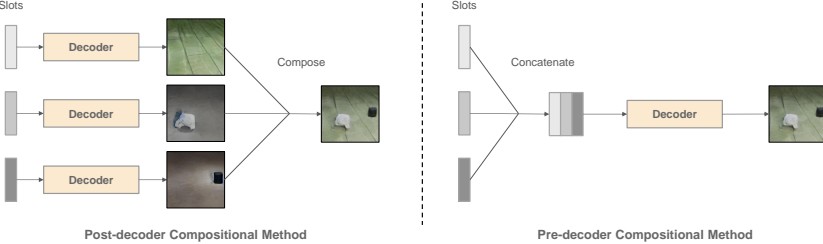

Figure 1: The overview of the post-decoder compositional method and pre-decoder compositional method. **Left**: Post-decoder compositional method follows the human intuition of first generating individual objects in the scene separately and then constructing the entire scene compositionally, but current post-decoder compositional methods are observed to struggle in complex scenes. **Right**: Recent pre-decoder compositional methods such as SLATE and LSD input all object representations (slots) together into a single decoder to construct the entire scene at once. They can handle complex scenes but are contrary to human intuition.

cused on exploring more powerful decoder architectures to enhance the ability of models to handle complex scenes. For example, SLATE (Singh et al., 2022a) and LSD (Singh et al., 2022b) introduce an autoregressive Transformer decoder and a diffusion decoder respectively to deal with real-world datasets. However, these pre-decoder compositional methods are different from the ways humans imagine or create scenes. In these methods, all object representations are concatenated together as one single representation and put into a single decoder to construct the entire scene at once. As a result, these methods fail to generate an image of a single object with its representation or edit single objects, limiting the generalization ability of models and leading to weak interpretability.

The limitation faced by current post-decoder compositional methods and pre-decoder compositional methods naturally leads to the question: ***can we design a model that is more intuitive to humans while maintaining its ability to handle complex scenes?*** We noticed that among the existing methods, the pre-decoder compositional methods that use a diffusion decoder can achieve better results in complex scenes. The denoising network adopted in these methods is pre-decoder compositional, which is still contrary to human intuition. However, we believe that the ability of these methods to handle complex scenes is mainly due to the diffusion process itself instead of the structure of the denoising network. Therefore, we naturally bright up the idea of introducing the post-decoder compositional modeling into the denoising network to make it more consistent with human intuition while still maintaining its ability to handle complex scenes. Specifically, this paper introduces a novel object-centric learning method, called **C**ompositional Scene Modeling with an **O**bject-centric **Di**ffusion **T**ransformer (CODiT). We design a **post-decoder** compositional diffusion architecture to denoise the latent, where we model the object's mask and construct the image latent in a compositional way. Each object representation will explicitly correspond to a specific area in the scene, and the noise in the corresponding area will be predicted. Finally, we will combine these noises at the spatial level. This modeling method is more consistent with human intuition of processing the entire scene by compositionally processing a single object than the past diffusion-based methods. Since our model follows the diffusion process, it retains the ability to handle complex scenes. We also discovered the theoretical explanation of CODiT from the perspective of Classifier-Free Guidance (CFG) (Ho & Salimans, 2022), which provides a new viewpoint for the research of human intuition as well as compositional scene modeling in the field of OCL.

Extensive experiments demonstrate that CODiT can not only adopt a single object representation to generate the corresponding image but also edit the objects by adjusting their masks during generation, which existing diffusion-based object-centric methods can hardly achieve. We also show that CODiT outperforms or competes with existing object-centric diffusion methods including LSD and SlotDiffusion on synthetic and real-world datasets in terms of segmentation and reconstruction.

In summary, the contributions of this work can be listed as follows:

1. CODiT is the first OCL model that introduces compositional modeling into denoising decoders, which is intuitive to humans and capable of handling complex scenes.

2. We explain the principles of CODiT from the perspective of CFG, which provides a new idea for human intuition as well as composition research in the field of OCL;

3. Experiments show that CODiT is the first diffusion-based OCL method to achieve single object generation and object editing tasks, which proves its strong interpretability.

4. Experiments show that the proposed CODiT outperforms existing diffusion-based OCL methods in segmentation and reconstruction tasks across multiple datasets.

## 2 RELATED WORK

Existing OCL methods can be mainly classified into two categories: post-decoder compositional methods and pre-decoder compositional methods. This classification is based on how they construct scenes——either by composing individual objects decoded from their respective representations or by concatenating the representations of all objects and inputting them into a decoder.

### 2.1 POST-DECODER COMPOSITIONAL METHODS.

Most early OCL methods are post-decoder compositional. N-EM (Greff et al., 2017) extracts the representations of objects through a neural EM algorithm. Tagger (Greff et al., 2016) adopts shared networks to update the appearance and mask of each object and sets denoising loss as the optimization target. RC  (Greff et al., 2015) fixes the mask of each object to $1/K$ and adopts pretrained DAE. CST-VAE (Huang & Murphy, 2015) iteratively infers the representation of each object by an RNN module. Compared with CST-VAE, AIR (Eslami et al., 2016) further predicts the existence of each object. SPAIR (Crawford & Pineau, 2019) segments the image into multiple regions to deal with scenes with more objects. MONet (Burgess et al., 2019) adopts component VAE for each object. For efficiency, IODINE (Greff et al., 2019) and SPACE (Lin et al., 2019) infer the object-centric representations in parallel. LDP (Yuan et al., 2019a) and GMIOO (Yuan et al., 2019b) focus on modeling the relation of occluded objects to deal with more complex multi-object scenes. GENESIS (Engelcke et al., 2019) uses autoregressive prior to modeling the relation between object shapes. GENESIS-V2 (Engelcke et al., 2021) and Slot Attention (Locatello et al., 2020) introduce arbitrary shape attention. The former presents the IC-SBP module to predict the shape attention of each object. The latter iteratively optimizes object-centric representations through cross-attention. Slot-VAE (Wang et al., 2023) improves Slot Attention through a hierarchical VAE. Similar to our work, these methods can model scenes in a post-decoder compositional way. However, they are only effective on simple synthetic scenes and cannot handle complex synthetic scenes. In contrast, our work outperforms these methods on complex synthetic scenes and real-world scenes.

### 2.2 PRE-DECODER COMPOSITIONAL METHODS.

To improve the model's ability to handle complex scenes, recent methods turn to construct scenes in a pre-decoder way: SLATE (Singh et al., 2022a) introduces an autoregressive Transformer decoder to reconstruct scene features instead of scene images. STEVE (Singh et al., 2022b) extends SLATE to video scenes by introducing a Transformer-based predictor to model the relation between the previous and next frames of the video. BO-QSA (Jia et al., 2022) learns the initial representations of objects and optimizes the Slot Attention model according to bi-level optimization. Similar to SLATE, BO-QSA adopts autoregressive Transformer decoders to deal with real-world datasets. To further improve the ability of the model, LSD (Jiang et al., 2023) and SlotDiffusion (Wu et al., 2023) introduce diffusion models into object-centric learning for the first time. What these methods have in common is that during the generation process, the representations of all objects are concatenated and then input into the diffusion model as a single condition. Although the above methods have proved to be effective in segmentation, they are contrary to human intuition to a certain degree. Unlike these methods, CODiT models object masks specifically in the diffusion-based decoder, which makes itself more intuitive to humans and has stronger interpretability.

### 2.3 OCL METHODS FOR OBJECT DISCOVERY

Compared with the earlier OCL methods, the recent methods focus more on the object discovery task or the binding problem. DINOSAUR (Seitzer et al., 2022) uses the pre-trained ViT model to extract scene features for learning object-centric representations, thereby significantly improving

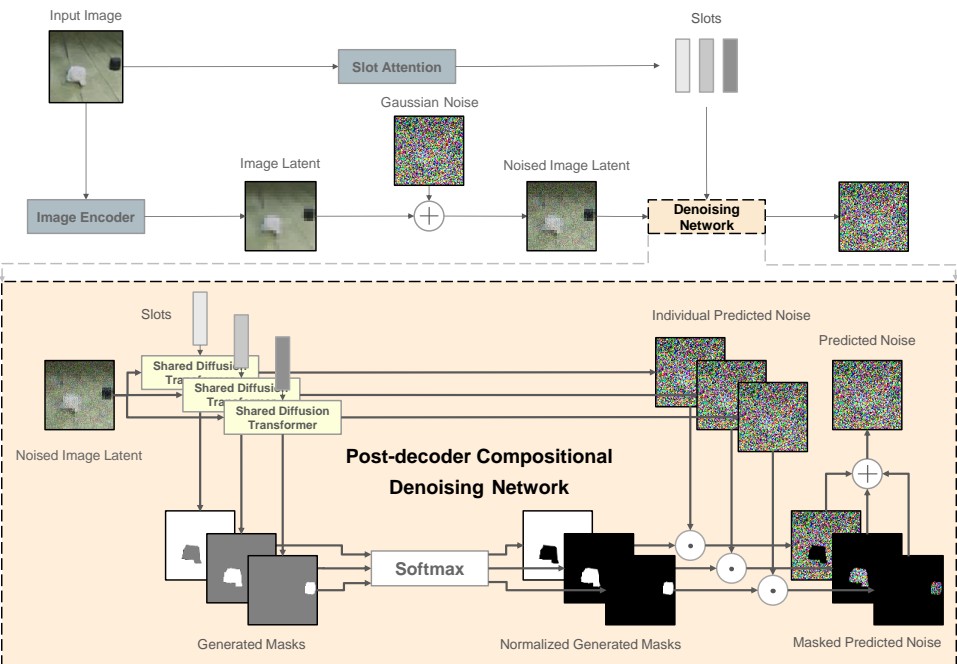

Figure 2: **Top**: The overview of CODiT. Similar to LSD and SlotDiffusion, we adopt the Slot Attention encoder to infer object slots. Then we adopt a pre-trained image encoder to extract the image latent and add noise to it. Finally, we input the noised image latent and object slots into the denoising network and predict noise. **Bottom**: During denoising, we adopt a post-decoder compositional denoising network, where each slot acts as a single condition to denoise the same noised image latent. The DiT decoder for denoising also computes object masks besides individual predicted noise. The model finally predicts noise by composing all of the individual predicted noise with normalized generated masks as weights.

the model's ability to segment complex scenes. CAE (Löwe et al., 2022) adopt complex values to represent objects, and Rotating Features (Löwe et al., 2024) modifies the complex features into the rotating features. OC-Net (Foo et al., 2024) computes the feature connectivity to discovery objects. Cyclic walks (Wang et al., 2024) extract object slots through the cyclic walks between the image features and slots. SPOT (Kakogeorgiou et al., 2024) adopts an autoregressive transformer decoder with sequence permutations and a teacher-student framework to improve the performance. VideoSAUR (Zadaianchuk et al., 2024) measures the temporal feature similarities to address real-world video. These methods have performed impressive results on complex real-world datasets. However, these methods can only segment the input scenes and cannot reconstruct or generate images. In comparison, the proposed method is equipped with generation ability.

## 2.4 COMPOSITONAL DIFFUSION MODELS.

Some recent studies (Liu et al., 2022; 2023; Su et al., 2024) focus on generating the entire image by compositional diffusion models. It should be clear that these methods can only produce images compositionally while they are unable to extract object representations from images. Rather than specific objects, the representations these models learn may reflect abstract ideas like color and light. Additionally, unlike most OCL methods, these methods are unable to provide object masks.

## 3 CODiT

CODiT mainly has three parts: a Slot Attention encoder for inferring object slots, an image encoder for extracting image latent where we add noise, and a post-decoder compositional denoising network for denoising image latent with object slots and reconstructing image pixels. The complete

model architecture can be seen in Figure 2. Our model follows the same diffusion process as the Latent Diffusion Model(LDM), which we first introduce. Then we will introduce the architecture of each part of CODiT, and finally demonstrate a theoretical explanation for CODiT from the view of classifier-free guidance (CFG).

## 3.1 PRELIMINARIES: LDM

The standard Latent Diffusion Model (LDM) (Rombach et al., 2022) first pre-trains an image autoencoder to extract image latent $z_0$ and then trains a conditional denoising diffusion model on the latent level. Following DDPM (Ho et al., 2020), the distribution of $z_0$ with the condition $S$ can be described as $p(z_0|S) = \int p(z_{0:T}|S)dz_{1:T}$. The joint distribution $p(z_{0:T}|S)$ is modeled as a Markov chain as $p(z_{0:T}|S) = p(z_T)\prod_{t=T,...,1} p(z_{t-1}|z_t, S)$, where $p(z_T) = \mathcal{N}(\mathbf{0}, I), p(z_{t-1}|z_t, S) = \mathcal{N}(\boldsymbol{\mu}_\theta(z_t, t, S), \beta_t I)$. $\beta_{1:t}$ is a increasing variance schedule and $\boldsymbol{\mu}_\theta$ is computed as $\boldsymbol{\mu}_\theta(z_t, t, S) = \frac{1}{\sqrt{\alpha_t}}(z_t - \frac{\beta_t}{\sqrt{1-\bar{\alpha}_t}}\hat{\boldsymbol{\epsilon}}_t)$, where $\alpha_t = 1 - \beta_t, \bar{\alpha}_t = \prod_{i=1}^t \alpha_t$, and $\hat{\boldsymbol{\epsilon}}_t = g_\theta(z_t, t, S)$ is the output of the diffusion decoder. Compared with LDM, the recent diffusion-based object-centric learning methods, including the proposed method, adopt object representations as the condition $S$.

During generation, LDM first samples a random standard Gaussian noise $z_T$ and samples $z_{T-1}, ..., z_0$ step by step according to $p(z_{t-1}|z_t, S)$ to finally get $z_0$, which can be used to reconstruct the image latent. For training, LDM first samples a timestep $t$ from $\{1, ..., T\}$ and a standard Gaussian noise $\boldsymbol{\epsilon}_t$. Then it gets noised image latent $z_t$ by $z_t = \sqrt{\bar{\alpha}_t}z_0 + \sqrt{1-\bar{\alpha}_t}\boldsymbol{\epsilon}$. The corresponding loss function can be described as $L = ||\boldsymbol{\epsilon}_t - g_\theta(z_t, t, S)||^2$.

## 3.2 SLOT ATTENTION ENCODER

The Slot Attention Encoder used in CODiT mainly follows the original Slot Attention. Given an input image $x \in R^{C \times H \times W}$, it will be first transformed into a feature map $x^{\text{feat}} \in R^{D_{\text{feat}} \times H_{\text{feat}} \times W_{\text{feat}}}$ through a backbone to extract the information from original images in different scales. We then extract $K$ object representations (slots) $s_i(i \in \{1, ..., K\})$ from the feature map $x^{\text{feat}}$. The range of $i$ will be omitted in the following description for clarity.

Specifically, We first initialize the set of object slots $S$ by sampling from a learnable Gaussian distribution. These object slots are then updated through a competitive cross-attention mechanism, where object slots provide queries and image feature map provides keys and values. All of the queries, keys, and values are computed through linear projection and have the same dimension $D$. We first apply dot-product on the queries and keys to get an $N \times K$ matrix, where $N$ stands for the number of inputs in $x^{\text{feat}}$. After dividing the matrix by $\sqrt{D}$, we adopt a softmax function along with the dimension $K$ to get attention map $A$. The attention map is then sum-pooled along with the dimension $N$ for each attention map $A_{n,m}$ ($1 \leq n \leq N, 1 \leq m \leq K$). Finally, we get the object slots for updation $S^{\text{upd}}$ by adding up all of the values $v(x^{\text{feat}})$ with the sum-pooled attention map $\hat{A}$ as weights. The process can be described as:

$$A = \text{softmax}_K\Big(\frac{q(S)k(x^{\text{feat}})^\top}{\sqrt{D}}\Big), \quad \hat{A}_{n,m} = \frac{A_{n,m}}{\sum_{l=1}^N A_{l,m}}, \quad S^{\text{upd}} = \hat{A}v(x^{\text{feat}}).$$

We then adopt a GRU module to update object slots as $S = f_\theta^{\text{GRU}}(S, S^{\text{upd}})$, where slots are then processed by the cross-attention mechanism and GRU for several rounds to provide final object slots.

## 3.3 IMAGE ENCODER

Earlier Object-Centric learning methods (such as Slot Attention and GENESIS-V2) try to reconstruct the input image while recent methods try to reconstruct the image latent (such as SLATE and DINOSAUR) or even to predict the noise added in the latent (such as SlotDiffusion or LSD). Following LDM, we first transform the input image into a latent and then train a diffusion denoising model on it. Given an image $x \in R^{C \times H \times W}$, we will first transform it into a latent $z_0 \in R^{C_{\text{AE}} \times H_{\text{AE}} \times W_{\text{AE}}}$ by a pre-trained LDM image encoder. During training, we will add noise to the latent and denoise it by the proposed Post-decoder compositional denoising network introduced in detail in the next section:

$$z_0 = f_{\text{img}}^{\text{enc}}(x).$$

### 3.4 Post-decoder Compositional Denoising Network

Since the object representations extracted capture information of image latent, it is convenient for us to adopt this information to help predict $z_0$ in the denoising network $g_\theta(z_t, t, S)$. Current conditional denoising networks force all of the conditions and noised latent to be input together into a single UNet, where all of the conditions guide the denoising process through a cross-attention mechanism with the feature map in certain layers of UNet. This design is suitable for language-conditioned scene generation tasks but is counterintuitive to humans as mentioned in Section 1, which leads to limitations in scene editing tasks that require object representation as a condition. For example, LSD cannot edit the appearances of specific objects in the scene (except for adding or erasing objects) via editing the representation of the corresponding object. These methods also cannot generate a single object based on the object representation like humans can, which demonstrates weak interpretability of the learned representations as well as the methods themselves. Due to the more complex modeling of competition between object representations in the cross-attention module of UNet, individual object representations do not need to have explicit meanings but can still predict the noise of the entire scene through this competition mechanism. In other words, the images decoded from these object representations will actually have significant differences from the images corresponding to these objects, which can be seen in Section 4.1.

By contrast, to fully exploit the information of each $s_i$ and make models be more intuitive to humans, we adopt a denoising decoder $g_\theta^{\text{dec}}$ shared across all object representations that revices only individual object representations $s_i^{\text{dec}}$ and the noised latent $z_t$ besides timestep $t$. Since only a single condition is given during the denoising process for each object, it is not suitable for us to introduce a cross-attention mechanism in the denoising decoder such as UNet. This is because a single condition will cause the corresponding attention map to be all ones and the information from image features of different levels will be diminished severely. To solve this problem as well as enable models to perform better, we adopt DiT (Peebles & Xie, 2023) as the backbone of our decoder. Specifically, the noise image latent is first patchified and copied for $K$ times. Each of them will then be denoised by the adaptive layer norm blocks with several self-attention layers in the DiT decoder, and they can be modified by scale and shift parameters computed from a single object slot. With this architecture, the input object slots are more like class conditions that can be fully exploited by networks instead of natural languages that have to compete with other object slots.

To integrate the denoising results of each object representation, $g_\theta^{\text{dec}}$ computes an extra unnormalized generation mask $m_i^{\text{gen}} \in [0, 1]^{1 \times H_{\text{AE}} \times W_{\text{AE}}}$ as well as predicted $\hat{\epsilon}_{t,i} \in R^{C_{\text{AE}} \times H_{\text{AE}} \times W_{\text{AE}}}$ for each object representation. Finally, we sum the predicted $\hat{\epsilon}_{t,i}$ with normalized weights $\hat{m}_i^{\text{gen}}$ and get $\hat{\epsilon}_{t,i}$:

$$\hat{\epsilon}_t = \sum_{i=1}^K \hat{m}_i^{\text{gen}} \hat{\epsilon}_{t,i}, \quad \hat{m}_i^{\text{gen}} = \text{softmax}_K(m_i^{\text{gen}}), \quad [m_i^{\text{gen}}, \hat{\epsilon}_{t,i}] = g_\theta^{\text{dec}}(z_t, \text{condition} = s_i, t).$$

Compared with previous methods, this way of modeling a single object and then compositionally constructing the entire scene is more in line with human intuition. Since we adopt the simple weighted summation module instead of the cross-attention mechanism to compute noise, the model will force each object representation to be decoded into more meaningful and complete objects. During the generation process, the reconstruction of the latent $\hat{z}_0 \in R^{C_{\text{AE}} \times H_{\text{AE}} \times W_{\text{AE}}}$ will be input into the pre-trained image decoder to get the reconstruction of whole image $\hat{x} \in R^{C \times H \times W}$.

As the generation mask $m_i^{\text{gen}}$ modifies the predicted noise in a similar way as classifier-free guidance (CFG), we also demonstrate a theoretical explanation for the proposed method from the view of CFG in Section 3.5, which shows the consistency between CODiT and current theory of diffusion model.

### 3.5 Insight from Classifier-Free Guidance

In Classifier-Free Guidance (CFG), the denoising model predicts noise both with and without condition, and uses the linear combination $\hat{\epsilon}_t(z_t, c) = \hat{\epsilon}_t(z_t, \emptyset) + w(\hat{\epsilon}_t(z_t, c) - \hat{\epsilon}_t(z_t, \emptyset))$ to compute noise. Given $K$ conditions, each condition $c_i$ has a corresponding guidance weight $w_i$, and the predicted image latent is computed as:

$$\hat{\epsilon}_t(z_t, c) = \hat{\epsilon}_t(z_t, \emptyset) + \Sigma_{i=1}^K w_i(\hat{\epsilon}_{t,i}(z_t, c_i) - \hat{\epsilon}_t(z_t, \emptyset)).$$

Now we replace $w_i$ and $\hat{\epsilon}_{t,i}(z_t, c_i)$ with the $\hat{m}_i^{\text{gen}}$ and $\hat{\epsilon}_{t,i}$ from our decoder. It should be noted that the original CFG weight $w_i$ is a scalar, which means we will apply the same weight at all positions.

However, as we replace it with the tensor $\hat{m}_i^{\text{gen}}$, this allows us to use different weights at different positions. Since $\Sigma_{i=1}^K \hat{m}_i^{\text{gen}} = 1$, we can compute the final predicted noise:

$$\hat{\epsilon}_t(z_t, c) = \hat{\epsilon}_t(z_t, \emptyset) + \Sigma_{i=1}^K \hat{m}_i^{\text{gen}}(\hat{\epsilon}_{t,i} - \hat{\epsilon}_t(z_t, \emptyset)) = \Sigma_{i=1}^K \hat{m}_i^{\text{gen}} \hat{\epsilon}_{t,i},$$

which is the same as the output of our post-decoder compositional denoising network. As we force the guidance weight to be normalized, there is no need to train an unconditional denoising model $\hat{\epsilon}_t(z_t, \emptyset)$. It should be pointed out that the CFG parameters $\hat{m}_i^{\text{gen}}$ here is a tensor of the same size as $\hat{\epsilon}_t$ instead of a scalar, which means that the predicted noise can be manipulated at a spatial level.

From a CFG perspective, CODiT can generate a meaningful image if we use only a few or even one representation $s_i$ as a condition during generation even if we use multiple representations during training. This is because the guidance weights are changeable to produce complete images with different control degrees during both the training and generation process. Current diffusion-based OCL methods, however, can only be seen as conditional diffusion models where the guidance is always 1. The CFG perspective can also be seen as a simulation of the process of human beings constructing scenes because humans can autonomously control the appearance of individual objects when constructing scenes without being completely constrained by the representation of the objects. CODiT can achieve this by manipulating the generated masks to control the generation process as long as they are normalized or adopting unnormalized masks together with an extra unconditional diffusion model. As a result, while maintaining segmentation and generation ability, CODiT can further edit objects and generate meaningful images with single object representation. We will prove this through experiments in Section 4.

## 4 EXPERIMENTS

In this section, we demonstrate the abilities of CODiT to edit and generate single objects, thereby verifying its strong interpretability. We also evaluate the segmentation and reconstruction performance on multiple datasets to evaluate the proposed method more comprehensively.

**Datasets.** Three synthetic datasets (i.e., CLEVRTEX (Karazija et al., 2021), MOVi-C and MOVi-E (Greff et al., 2022)) and three real-world datasets (i.e., OCTScenes-B (Huang et al., 2023), FFHQ (Karras et al., 2019) and PASCAL VOC 2012 (Everingham et al., 2010)) are used to evaluate the proposed method. CLEVRTEX is the complicated version of CLEVR (Johnson et al., 2017) with complex textures of objects and backgrounds. MOVi-C is a video scene dataset, and the sample is processed as individual images in our experiments. Compared with CLEVRTEX, MOVi-C has more complex objects and natural backgrounds. MOVi-E contains more objects compared with MOVi-C. OCTScenes-B (OCT-B) has multi-view scenes with static objects placed on a table. The scenes in the datasets are treated as individual images like MOVi-C. FFHQ contains images of human faces with similar layouts, which is suitable for testing the generation ability of object-centric learning methods. VOC is a real-world dataset commonly used in object detection and segmentation. It has recently been used in object-centric learning methods to measure the performance of complex natural datasets. Like DINOSAUR, the image size of PASCAL VOC 2012 (VOC) is set to $224 \times 224$, while that of other datasets is set to $256 \times 256$.

**Baselines.** Since current post-decoder methods have been proven to struggle with complex datasets (Karazija et al., 2021), we mainly select two diffusion-based methods (i.e., LSD (Jiang et al., 2023) and SlotDiffusion (Wu et al., 2023) (SD) as the baselines to demonstrate the superiority of the proposed compositional diffusion decoder. Similar to LSD, we also use a pre-trained image autoencoder to get the image feature map. By contrast, SlotDiffusion trains an individual image encoder for each dataset, and we use image-based SlotDiffusion for fair comparison. We also compared with DINOSAUR (Seitzer et al., 2022) and BO-QSA (Jia et al., 2022) in Appendix E.6 and Appendix E.2 respectively to report the performance of CODiT in real-world complex datasets as well as the possibility of combining CODiT with other modified Slot Attention encoder.

### 4.1 THE INTERPRETABILITY OF OBJECT REPRESENTATIONS

This section exhibits the interpretability of object representations extracted by CODiT as well as the compared methods via visualizing individual slots through the single-condition generation process. We evaluate the performance of CODiT, LSD, and SlotDiffusion on CLEVRTEX and FFHQ

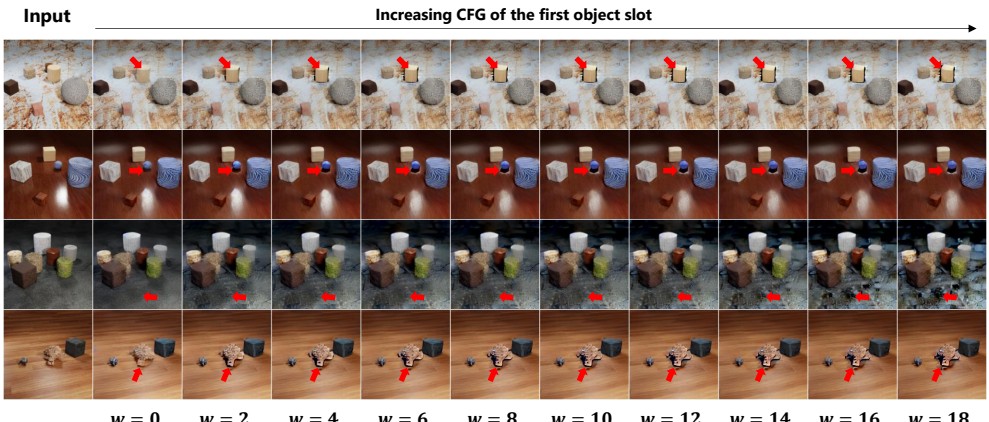

Figure 3: Individual object generation results. The first row of each figure represents the complete input image and the masked input image, while the second row represents the reconstruction image and the generation results that are conditioned on the individual slot corresponding with the mask.

**Input** — Increasing CFG of the first object slot

$w = 0$    $w = 2$    $w = 4$    $w = 6$    $w = 8$    $w = 10$    $w = 12$    $w = 14$    $w = 16$    $w = 18$

Figure 4: The visualization of object editing experiments. The weight tensor $w$ varies from 0 to 18 from the second column to the last column. As the weight tensor (also the CFG parameter of the first object slot) increases, the contrast degree of corresponding objects becomes higher.

datasets. As shown in Figure 3, CODiT can generate the corresponding targets with a single slot. An interesting result is that even if the generation masks are nearly blank for some object slots, they can still generate meaningful images. This implies that their masks are blank only because their generation may be helpless during reconstruction. For FFHQ, the proposed method can also generate meaningful parts (e.g., hair, faces, and backgrounds) given corresponding slots. The visualization results prove that the slots extracted by CODiT have strong interpretability.

In comparison, LSD and SlotDiffusion struggle to generate the corresponding image with a single object slot unless it corresponds to the background. They generate complete scenes (for LSD) or multiple objects (for SlotDiffusion) even if only one object slot is given. We attribute this phenomenon to the fact that both LSD and SlotDiffusion force object and background slots to compete through the cross-attention mechanism during training. As a result, when the background representation is omitted, a single object slot can not locate itself in the whole image, which may cause the model to generate multiple objects. On the other hand, as the denoising networks in these methods predict the complete image noise directly, they still tend to generate complete scenes given only a single slot, which brings poor interpretability to these slots. By contrast, the proposed post-decoder compositional diffusion architecture forces slots to extract more information about single objects and therefore have stronger interpretability.

## 4.2 OBJECT EDITING

Although LSD and SlotDiffusion have shown that they can edit images by selecting certain slots, they can not edit the object appearances since they lack modeling object masks (or CFG parameters)

Table 1: Comparison results of unsupervised segmentation. Since current post-decoder methods have been proven to struggle with complex datasets, we select two diffusion-based methods for comparison. We adopt the attention masks of LSD and SlotDiffusion, and the generated masks of CODiT. All masks are upsampled to the image size before computing metrics.

| Dataset | Model | ARI-A↑ | ARI-O↑ | AMI-A↑ | AMI-O↑ | mIOU↑ | mBO↑ |
|---------|-------|--------|--------|--------|--------|-------|------|
| CLEVRTEX | LSD | **79.78** | 68.51 | 68.16 | 75.39 | 58.90 | 65.28 |
|          | SD | 13.67 | 68.88 | 37.46 | 72.25 | 55.18 | 54.41 |
|          | CODiT | 77.48 | **90.81** | **70.44** | **91.54** | **62.37** | **67.82** |
| OCT-B | LSD | 30.49 | 62.85 | 41.75 | 75.24 | 35.58 | 38.61 |
|       | SD | 10.14 | 41.47 | 41.97 | 60.86 | 43.30 | 42.51 |
|       | CODiT | **74.96** | **79.29** | **68.40** | **82.50** | **51.63** | **56.60** |
| MOVi-C | LSD | 41.06 | 52.76 | 45.34 | 63.18 | 40.30 | 46.69 |
|        | SD | 11.66 | **53.42** | 31.28 | 63.65 | 34.32 | 34.45 |
|        | CODiT | **47.80** | 52.91 | **49.41** | **64.82** | **40.79** | **47.80** |
| MOVi-E | LSD | 47.64 | 47.12 | 51.03 | 70.41 | 35.19 | 39.60 |
|        | SD | 9.65 | **54.50** | 39.88 | **73.74** | **35.63** | 36.05 |
|        | CODiT | **50.09** | 47.91 | **52.73** | 71.48 | 35.58 | **39.77** |

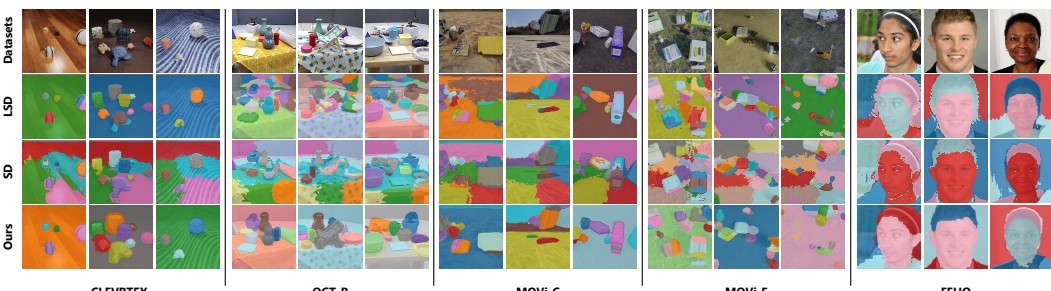

Figure 5: The segmentation results of LSD, SlotDiffusion, and CODiT.

during the generating process. By contrast, CODiT can edit object appearances by adjusting the CFG parameters of certain objects, which has been discussed in Section 3.5. The corresponding results can be seen in Figure 4

We trained an extra unconditional diffusion model to edit object appearance. During generation, we adopt both the conditional and unconditional diffusion models and follow standard CFG to produce images. To edit the appearance of a single object, after we extract the generated masks and normalize them, we further add a certain weight tensor $w$ to the generated mask of the first slot $\hat{m}_1^{\text{gen}}$. Additionally, we zero-pad the places of $w$ where the weights of $\hat{m}_1^{\text{gen}}$ is not the largest one among $\hat{m}_{1:K}^{\text{gen}}$. These masks will then act as CFG parameters during the generation process. The results show that, as the tensor $w$ increases, the contrast degree of corresponding objects or backgrounds becomes higher. Specifically, the edited objects (the white cylinder in the first row, the blue sphere in the second row, and the brown monkey head in the fourth row) are more obvious than other objects. The shadow beneath these objects and the light shed on them are also more clear. For the third row, it is noticeable that the appearance of the background has changed. It's striking that the model hardly edits other objects, which means that We achieve object-centric and region-specific CFG guidance.

## 4.3 UNSUPERVISED SEGMENTATION

This section will evaluate the unsupervised segmentation performance of the proposed method. Adjusted Rand Index(ARI) (Cugmas & Ferligoj, 2015) and Adjusted Mutual Information(AMI) (Vinh et al., 2010) are two main metrics of segmentation reported in the quantitative result. Specifically, we use two variants for each metric to evaluate the segmentation performance more thoroughly: ARI-A and AMI-A are calculated by considering all of the pixels, while ARI-O and AMI-O are calculated by only considering the pixels of the foreground pixels. In addition, we also report the mIOU and mBO for these methods. The attention masks inferred by LSD and SlotDiffusion, as well as the generated mask output by the denoising networks in CODiT, are upsampled to the image size for both computation and visualization. The results can be seen in Table 1 and Figure 5.

Table 2: Comparison results of image reconstruction.

|  | Model | CLEVRTEX | OCT-B | MOVi-C | MOVi-E |
|---|---|---|---|---|---|
| MSE | LSD | 1.81e-2 | 3.22e-2 | 1.56e-2 | 1.93e-2 |
|  | SD | 1.62e-2 | 3.14e-2 | 1.64e-2 | 2.23e-2 |
|  | CODiT | **1.48e-2** | **2.75e-2** | **1.40e-2** | **1.65e-2** |
| LPIPS | LSD | 0.267 | 0.225 | 0.334 | **0.306** |
|  | SD | 0.241 | **0.213** | 0.354 | 0.347 |
|  | CODiT | **0.225** | 0.223 | **0.304** | 0.315 |

It can be seen that CODiT achieves better or comparable segmentation performance compared with LSD and SlotDiffusion among all of the datasets. Although recent methods use ARI-O and AMI-O to measure their segmentation ability, we found them inappropriate, which is also reported in the LSD paper. As shown in Figure 5, although SlotDiffusion has comparable or better ARI-O and AMI-O metrics, the visualization results show that it still tends to segment the background into more than one piece. To better measure the segmentation quality of the proposed method and baselines, we propose introducing ARI-A and AMI-A. The result shows that SlotDiffuion struggles in these two metrics while the proposed method performs the best among the three methods except for the ARI-A of CLEVRTEX, which is lower than LSD. As shown in Figure 5, LSD has good background segmentation capabilities on the CLEVRTEX dataset but may over-segment small foreground objects. Since the background part of CLEVRTEX that occupies most of the area in the image plays a decisive role in calculating ARI-A, the segmentation effect on the foreground object will be weakened. SlotDiffusion also has a higher mIOU than CODiT on the MOVi-E dataset, which we attribute to the larger number of objects on MOVi-E. When there are enough objects in the scene, the area occupied by the background will be significantly reduced, so the defect of SlotDiffusion tending to divide the background into multiple parts will be compensated. It is worth noticing that the proposed method achieves the highest ARI-A and ARI-O on more complex datasets compared with other baselines, which indicates the potential of CODiT in complex datasets. This can also be proved by the visualization on FFHQ, where CODiT demonstrates more precise semantic segmentation. By contrast, the segmentation results of LSD and SlotDiffusion at the junction of various parts are quite different from human intuition.

### 4.4 Image Reconstruction

We also have a simple insight into the reconstruction ability of CODiT as well as baselines in Table 2 in terms of MSE and LPIPS, where CODiT demonstrates the best results among all of the four datasets in MSE and achieves the best result in two datasets in terms of LPIPS. We notice that LSD performs worse in relatively simple datasets such as CLEVRTEX and OCT-B, which we attribute to the overfitting phenomenon, which is also mentioned in the original paper of LSD. As the cross-attention mechanism in the decoder of LSD plays an important role in composing slots during denoising, the slots themselves may not obtain enough information about objects. As a result, during reconstructing images from test datasets where the combination of objects is relatively novel for the cross-attention mechanism, the performance on reconstruction will be worse. By contrast, with the proposed post-decoder compositional denoising network, CODiT can extract representations with more information as well as stronger interpretability and therefore achieves better or comparable reconstruction results compared with baselines.

## 5 Conclusion

In this paper, we introduce CODiT, a novel object-centric learning method with a post-decoder compositional diffusion network that can predict noise in a compositional way. We also give a preliminary theoretical explanation of our method from a Classifier-Free Guidance Perspective. Experiments on several datasets show that CODiT can generate individual objects with a single object slot and edit objects in the scenes, which demonstrates the stronger interpretability of CODiT that other baselines can seldom achieve. We also demonstrate that CODiT can achieve better or comparable results compared with the SOTA method in terms of segmentation and reconstruction.

## REPRODUCIBILITY STATEMENT

We have placed the model's code in the **Supplementary Material** and provided implementation details of the main metrics (Appendix D.1), datasets (Appendix D.2), baselines (Appendix D.3), and the model structure of the proposed method (Appendix D.4). We believe this will contribute to improving the reproducibility of our work.

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

## A    IMPACT STATEMENTS

Since our method performs a generation task, it may generate objectionable or biased content. Similar to other image editing or image generation models, specific malicious uses faced by our model may include editing faces to achieve identity forgery or posting certain generated weird images on the Internet to cause panic in the community. Future research should avoid these malicious uses.

## B    LIMITATIONS AND FUTURE WORKS

Although CODiT has shown impressive results on multiple tasks, the DiT decoder of the proposed method has more parameters than the UNet decoders of similar methods, which increases the space required to store the model. This is discussed in detail in Appendix E.4. On the other hand, while we have explored the controllable compositional generation capability in Appendix E.3, the generation performance of CODiT on more complex real-world datasets (such as VOC) remains an area worthy of further investigation, as it continues to pose a challenging problem in the field of unsupervised learning. Fortunately, as shown in Appendix E.6, the proposed method has shown promising segmentation results on real-world datasets.

Another limitation of CODiT is that CODiT does not model the relations between slots as precise as LSD and SlotDiffusion, as a result, it may cause unnatural artifacts during composing slots from different images. We think the balance between the reality and the interpretability is reasonable. From a subjective perspective, as illustrated in Figure E.3, we think the shown compositional generation results are acceptable. This is mainly because the pre-trained image autoencoder can capture and erase these artifacts to a certain extent. However, we think this is still an interesting and meaningful area for future research.

The next step of our work may be exploring compositional generating in more natural datasets. Although some of the current methods have tried learning object representations in natural datasets, they may not have the generation ability or they do not generate images in a compositional way. Exploring the possibility of extracting object representations without supervision and adopting these representations to generate images compositionally is still a challenging but meaningful research field. Another meaningful research area is the unsupervised extraction of the properties(e.g., position and size) of objects. With these disentangled representations, there may be more approaches to control the generation process with the aid of CFG. Current methods (Jiang & Ahn, 2020; Wang et al., 2023; Wu et al.) has shown promising results on learning these disentangled representations on relatively simple datasets such as CLEVR, it will be an important part for our future work to extent these method into more complex datasets or even new world.

## C    ABLATION STUDY

We show the importance of the DiT architecture and the design of the post-decoder compositional module in this section. To show the importance of the DiT architecture, we replace the DiT decoder with the UNet decoder without a cross-attention mechanism where object slots act as class embeddings. This is because we only input a single slot into the decoder and the cross-attention mechanism in the UNet will accordingly fail. We also tried the DiT block with cross-attention instead of adaLN-Zero, where all of the slots are input into one decoder, to prove the effectiveness of the post-decoder

Table 3: Results of ablation study. 'UNet Decoder' represents the model whose DiT decoder is replaced by a UNet decoder, 'DiT-CA Decoder' represents the model whose DiT decoder with adaLN-Zero is replaced by the DiT decoder with cross-attention, and 'Full' represents the complete CODiT. AS 'DiT-CA Decoder' does not model masks during generation, we use the attention mask instead to evaluate the segmentation performance.

| Dataset | Model | ARI-A↑ | ARI-O↑ | AMI-A↑ | AMI-O↑ | mIOU↑ | mBO↑ |
|---------|-------|--------|--------|--------|--------|-------|------|
| CLEVRTEX | UNet Decoder | 4.21 | 15.12 | 15.82 | 26.03 | 8.04 | 13.80 |
| | DiT-CA Decoder | 24.70 | 50.06 | 42.50 | 62.01 | 54.03 | 54.61 |
| | Full | **77.48** | **90.81** | **70.44** | **91.54** | **62.37** | **67.82** |
| OCT-B | UNet Decoder | 25.55 | 29.58 | 29.81 | 41.76 | 9.35 | 12.56 |
| | DiT-CA Decoder | 27.87 | 69.58 | 42.28 | 78.25 | 36.59 | 38.96 |
| | Full | **74.96** | **79.29** | **68.40** | **82.50** | **51.63** | **56.60** |
| MOVi-C | UNet Decoder | 19.41 | **56.92** | 34.06 | **67.66** | 33.96 | 36.58 |
| | DiT-CA Decoder | 40.46 | 47.53 | 43.57 | 58.60 | **42.78** | 47.65 |
| | Full | **47.80** | 52.91 | **49.41** | 64.82 | 40.79 | **47.80** |
| MOVi-E | UNet Decoder | 12.30 | **54.61** | 38.94 | **73.89** | 30.90 | 31.85 |
| | DiT-CA Decoder | 48.64 | 48.43 | 52.69 | 69.45 | **37.84** | **42.03** |
| | Full | **50.09** | 47.91 | **52.73** | 71.48 | 35.58 | 39.77 |

compositional module. The results can be seen in Table 3, where 'UNet Decoder' represents the model whose DiT decoder is replaced by a UNet decoder, 'DiT-CA Decoder' represents the model whose DiT decoder with adaLN-Zero is replaced by the DiT decoder with cross-attention, and 'Full' represents the complete CODiT. AS 'DiT-CA Decoder' does not model masks during generation, we use the attention mask instead to evaluate the segmentation performance.

The quantitative results, especially ARI-A and AMI-A, show that the DiT architecture and the post-decoder compositional module can improve the performance of segmentation. The model with the 'UNet Decoder' fails in more simple datasets such as CLEVRTEX and OCT-B, which is also reported in the original paper of LSD. On the contrary, the DiT decoder is suitable for datasets with different levels of difficulty. It is also noticeable that the 'DiT-CA Decoder' performs comparably with the proposed method on MOVi-E, the most complex dataset among the four datasets, which shows its potential in more complex datasets. However, it should be noticed that it does not adopt the compositional module, which leads to similar problems as described in Section 1.

# D  IMPLEMENTATION DETAILS

## D.1  DETAILS OF METRICS

**ARI.** We suppose the image has $N = H \times W$ pixels. Given the ground-truth segmentation map of an image $m \in \{1, ...a, K\}^N$ and predicted segmentation map $\hat{m} \in \{1, ..., \hat{K}\}^N$, where $K$ and $\hat{K}$ represent the ground-truth number of objects and the estimated number of objects respectively, we first transform them into one-hot vectors $r \in [0, 1]^{K \times N}$ and $\hat{r} \in [0, 1]^{\hat{K} \times N}$. Then we select all of the pixels in $r$ and $\hat{r}$ for ARI-A or that belong to foreground objects in the ground-truth segmentation map for ARI-O to get $r^{sel} \in [0, 1]^{K \times D}$ and $\hat{r}^{sel} \in [0, 1]^{\hat{K} \times D}$. $D$ stands for the number of ground-truth selected pixels. Then we compute the following intermediate variables:

$$b_{all} = \sum_{i=1}^{K} \sum_{j=1}^{\hat{K}} C(t_{i,j}, 2)$$

$$b_{row} = \sum_{i=1}^{K} C(\sum_{j=1}^{\hat{K}} t_{i,j}, 2))$$

$$b_{col} = \sum_{j=1}^{\hat{K}} C(\sum_{i=1}^{K} t_{i,j}, 2))$$

$$c = C(\sum_{i=1}^{K} \sum_{d \in D} r_{i,d}^{sel}, 2),$$

where $C(x, y)$ represents the number of combinations $\frac{x!}{(x-y)!y!}$ and $t_{i,j}$ represents the dot product $\sum_{d \in D} r_{i,d}^{sel} \cdot \hat{r}_{j,d}^{sel}$. Finally, we compute ARI as:

$$\text{ARI} = \frac{b_{all} - b_{row}b_{col}/c}{(b_{row} + b_{col})/2 + b_{row}b_{col}/c} \times 100$$

where we multiply the final result by 100 compared with the original results for presentation simplicity.

**AMI.** Suppose the test sets have $I$ visual scenes. let $\hat{K}_i$ be the true maximum number of objects appearing in the $i$th visual scene. and let $K_i$ be the estimated maximum number of objects appearing in the $i$th visual scene. Note that $\hat{K}_i$ and $K_i$ are not necessarily equal. $\hat{r}_i \in \{0,1\}^{(\hat{K}_i+1) \times N}$ and $r_i \in \{0,1\}^{(K_i+1) \times N}$ respectively represent the true and estimated one-hot vector of the $i$th scene corresponding to the pixel-wise partitions (including the foreground and background). $\mathcal{D}^i$ denotes the index sets that belong to the object areas in the $i$th scene, i.e., $\mathcal{D}^i = \{n \mid x_n^i \in$ object areas$\}$. Let $\hat{U}_k^i$ be the real index sets w.r.t. object $k$ in the $i$th scene, i.e., $\hat{U}_k^i = \{n \mid x_n^i \in$ areas of object $k\}$ $(0 \leq k \leq \hat{K}_i)$. Let $U_k^i$ be the estimated index sets w.r.t. object $k$ in the $i$th scene. $\hat{U}_k^i = \{n \mid \hat{x}_n^i \in$ areas of object $k\}$ $(0 \leq k \leq \hat{K}_i)$, where $\hat{x}$ is the reconstructed image. Let $\hat{m}^i \in [0,1]^{\hat{K}_i \times N}$ and $m^i \in [0,1]^{K_i \times N}$ be the true and estimated pixel-wise masks that indicate the object(including the background) weight for each pixel in each viewpoint. Let $\hat{a}^i \in [0,1]^{\hat{K}_i \times N \times 3}$ and $a^i \in [0,1]^{K_i \times N \times 3}$ be the true and estimated appearance of objects in the $i$th scene.

The computation of Adjusted Mutual Information (AMI) is described as:

$$\text{AMI} = \frac{1}{I} \sum_{i=1}^{I} \frac{\text{MI}(\hat{l}^i, l^i) - \mathbb{E}[\text{MI}(\hat{l}^i, l^i)]}{(\text{H}(\hat{l}^i) + \text{H}(l^i))/2 - \mathbb{E}[\text{MI}(\hat{l}^i, l^i)]} \times 100$$

where $\hat{l}^i \in \mathbb{R}^{\hat{K}_i+1}$ and we multiply the final result by 100 compared with the original results for presentation simplicity. $\hat{l}^i$ denotes the true probability distribution of the $i$th visual scene, i.e., $\hat{l}^i = \{|\hat{U}_k|/|\mathcal{D}^i| \mid 0 \leq k \leq \hat{K}_i\}$. $l^i$ is the estimated probability distribution, i.e., $l^i = \{|U_k|/|\mathcal{D}^i| \mid 0 \leq k \leq K_i\}$. H and MI respectively represent the entropy and mutual information of the distribution and their mathematical forms are described as:

$$\text{H}(\hat{l}^i) = -\sum_{k=0}^{\hat{K}_i} \hat{l}_k^i \log \hat{l}_k^i$$

$$\text{H}(l^i) = -\sum_{k=0}^{K_i} \sum l_k^i \log l_k^i$$

$$\text{MI}(\hat{l}^i, l^i) = \sum_{m=0}^{\hat{K}_i} \sum_{n=0}^{K_i} p_{m,n}^i \log \left( \frac{p_{m,n}^i}{\hat{l}_m^i \cdot l_n^i} \right)$$

where $\hat{l}_k^i$ and $l_k^i$ respectively note the true and estimated probability that the pixel in the $i$th image is partitioned to object $k$. $p_{m,n}^i$ denotes the probability w.r.t. pixels in the $i$th scene are divided into objects $m$ in the first set and objects $n$ in the second set. $p_{m,n}^i$ is calculated as follows:

$$p_{m,n}^i = \frac{o_{m,n}^i}{|\mathcal{D}^i|} = \frac{|\hat{U}_m^i \cap U_n^i|}{|\mathcal{D}^i|}$$

The matrix $\boldsymbol{o}^i \in \mathbb{R}^{(\hat{K}_i+1)\times(K_i+1)}$ is called the contingency table. And the expectation of MI can be analytically computed:

$$\mathbb{E}\big[\mathrm{MI}(\hat{\boldsymbol{l}}^i, \boldsymbol{l}^i)\big] = \sum_{m=0}^{\hat{K}_i} \sum_{n=0}^{K_i} \sum_{k=(a_m^i+b_n^i-N)^+}^{\min(a_m^i, b_n^i)} \frac{k}{N} \cdot \log\left(\frac{N \times k}{a_m^i \times b_n^i}\right)$$

$$\frac{a_m^i! b_n^i! (N-a_m^i)! (N-b_n^i)!}{N! k! (a_m^i - k)! (b_n^i - k)! (N - a_m^i - b_n^i + k)!}$$

where $(a_m^i + b_n^i - N)^+ = \max(1, a_m^i + b_n^i - N)$, $a_m^i$ and $b_n^i$ respectively represent the sum of rows and columns w.r.t. $\boldsymbol{o}^i$:

$$a_m^i = \sum_{n=0}^{K_i} o_{m,n}^i, \quad b_n^i = \sum_{m=0}^{\hat{K}_i} o_{m,n}^i$$

When calculating AMI-O, we will only consider pixels belonging to the foreground, while AMI-A needs to consider all pixels.

**mIOU&mBO.** Similar to ARI, after we get $r \in [0,1]^{K \times N}$ and $\hat{r} \in [0,1]^{\hat{K} \times N}$, we compute match index $\xi = \underset{\xi \in \Xi}{\arg\max} \sum_{i=1}^K \sum_{d \in D}(r_{i,d}^{fg} \cdot \hat{r}_{\xi_i,d}^{fg})$ according to the Hungarian algorithm Kuhn, where $\Xi$ represents total $\frac{\hat{K}!}{(\hat{K}-K)!}$ possible match ways. Finally, we compute the mIOU as:

$$\mathrm{mIOU} = \frac{1}{K} \sum_{k=1}^K \frac{\sum_{n=1}^N \min(r_{k,n}, r_{\xi_k,n})}{\sum_{n=1}^N \max(r_{k,n}, r_{\xi_k,n})} \times 100,$$

where we multiply the final result by 100 compared with the original results for presentation simplicity.

For mBO, we simply assign the ground-truth segment to the slot with the largest overlap, where there is no strict one-to-one mapping between the ground-truth and predicted masks.

**IS&FID.** Given a distribution of images $p(x)$, we compute the IS as:

$$\mathrm{IS} = \exp(E_{x \sim p(x)}(KL(p(y|x)||p(y)))),$$

where $KL(\cdot||\cdot)$ stands for the KL divergence between two distributions. we get $p(y|x)$ by input the image into a pre-trained Inception Net-V3 Szegedy et al. (2016). During measuring, we use every 10 generated images as a batch to compute $p(y|x)$ and get corresponding $p(y)$ by averaging the 10 $p(y|x)$, then we compute the mean IS of each batch accordingly. We average on IS of all batches to get the final IS.

Fréchet Inception Distance (FID) is a metric used to evaluate the quality of generated images, measuring similarity by comparing the feature distributions of generated and real images. The calculation involves extracting features from real and generated images using a pre-trained Inception network, then computing their means $\mu_X, \mu_Y$ and covariances $\Sigma_X, \Sigma_Y$. Finally, the FID value is derived using the formula:

$$\mathrm{FID} = \|\mu_X - \mu_Y\|^2 + \mathrm{Tr}(\Sigma_X + \Sigma_Y - 2\sqrt{\Sigma_X \Sigma_Y}),$$

where $\|\mu_X - \mu_Y\|^2$ is the squared distance between the means, $\mathrm{Tr}(\cdot)$ is the trace of the matrix, and $\sqrt{\Sigma_X \Sigma_Y}$ is the square root of the covariance matrices. A smaller FID value indicates that the distribution of generated images is closer to that of real images in the feature space, reflecting higher quality.

### D.2 DETAILS OF DATASETS

We use the official original version of CLEVRTEX. As there is no official split for CLEVRTEX, the first 48000 images are used to train and the trained model is tested on the last 1000 images. During training, we first resize the smaller dimension of images into 256 and then perform random

cropping to 256×256, during testing, we perform center cropping instead. For MOVi-C and MOVi-E, we trained our model on the train set and tested them on the validation set as the official test datasets are made for out-of-distribution(OOD) tasks. For OCT-B, We use the official split for training and testing. Following the implementation of the original paper, we divided each scene into 10 sub-scenes and randomly selected 3 images of sub-scenes in each training epoch. During testing, we select the first 3 images of each sub-scene as test datasets. For FFHQ, we treat the first 60000 images as a training split and the last 5000 split as a testing split. For VOC, we train all models on the 'trainaug' variant with 10582 images and test on the validation set with 1449 images. We first resize the smaller dimension of images into 224 and then perform center cropping to 224×224 during training and testing.

### D.3    DETAILS OF BASELINES

**LSD.** We used the official implementation of LSD[1]. Models for OCT-B were trained with hyperparameters similar to the one described in the original LSD paper for CLEVRTEX with the difference that(1) the number of slots is 15, (2) the size of output channels and slot is 192, and (3)the depth of the denoising UNet is 4. Models for MOVi-C/MOVi-E/CLEVRTEX/FFHQ were trained with hyperparameters the same as the one described in the original LSD paper for corresponding datasets.

**SlotDiffusion.** We used the official implementation of SlotDiffusion[2]. All of the Image Auto-Encoders were trained with default parameters described in 'slotdiffusion/img_based/configs/sa_ldm /vqvae_clevrtex_params-res128.py' with the differences that(1)the resolution was changed to (256,256), (2)the learning rate was changed to 1e-4, (3)the base channel was changed to 128, and(4) the channel multipliers was changed to [1,2,4,4]. We performed the last two changes so that SlotDiffusion can be run in the datasets of resolution 256×256.

For the SlotDiffusion module, all models were trained with default parameters described in 'slotdiffusion/img_based/configs/sa_ldm /vqvae_clevrtex_params-res128.py' with the difference that the base learning rate was changed to 5e-5 for CLEVRTE and OCT-B instead of 1e-4.

**BO-QSA.** We used the official implementation of BO-QSA[3]. The model was trained with default parameters described in 'BO-QSAtrain/train_trans_dec.py' with the differences that(1)the resolution was changed to (256,256), (2)the decoder utilized the Autoregressive Transformer Decoder, and (3)the batch size is 4, and(4) the number of samples is 4.

The hyperparameters used in model training on the data sets OCT-B and CLEVRTEX are the same, except that the number of slots on OCT-B is 15, and the number of slots on CLEVRTEX is 11.

### D.4    MODEL ARCHITECTURE AND HYPERPARAMETERS

We divide CODiT into three parts as described in Section 3 and we will introduce the implementation details of them in this section. The complete model architecture as well as hyperparameters is listed in Table 4. During training, we modify the learning rate by both linear warmup and exponential decay.

For the SlotAttention Encoder, we further divide it into two parts: *UNet Backbone* and *Slot Attention(SA)*. The *UNet Backbone* transforms the input image into an image feature map. Then we process the *Slot Attention* module to get object slots.

We also describe the implementation of *Image Auto-Encoder(AE)* and $g_\theta^{DiT}$ as *DiT*. After we get the image latents, they will be scaled to be processed by subsequent modules during training. In the generation process, after we get the predicted image latent, it will also be scaled back and input to the Image Decoder to get the final image. The scaling size is shown as *Image Scaling*.

## E    SUPPLYMENTARY EXPERIMENTS AND EVALUATION

---

[1] https://github.com/JindongJiang/latent-slot-diffusion

[2] https://github.com/Wuziyi616/SlotDiffusion

[3] https://github.com/YuLiu-LY/BO-QSA

Table 4: configuration of CODiT.

| | Hyperparameter | Datasets | | | | |
|---|---|---|---|---|---|---|
| | | CLEVRTEX | OCT-B | MOVi-C/E | FFHQ | VOC/COCO |
| **General** | Batch Size | 8 | 48 | 8/4 | 12 | 12 |
| | Training Epochs | 20 | 15 | 5/10 | 25 | 50 |
| | #Slots | 11 | 15 | 11/24 | 4 | 6/7 |
| | K-means Clusters | 5 | 9 | 18 | 4 | N/A |
| **UNet Backbone** | Input Resolution | 256 | | | | 224 |
| | Output Resolution | 64 | | | | 56 |
| | Base Channels | 128 | | | | |
| | Channel Multipliers | [1,2,2,4] | | | | |
| | Self Attention | Middle Blocks | | | | |
| | # Res Blocks | 2 | | | | |
| | # Attention Heads | 8 | | | | |
| | Output Channels | 128 | | | | |
| | Learning rate | 3e-5 | | | | |
| **SA** | Input Resolution | 64 | | | | 56 |
| | # Iterations | 3 | | | | |
| | Slot Size | 768 | 384 | 768 | 768 | 1152 |
| | Learning rate | 3e-5 | | | | |
| **AE** | Model | KL-8 | | | | |
| | Input Resolution | 256 | | | | 224 |
| | Output Resolution | 32 | | | | 28 |
| | Output Channels | 4 | | | | |
| **DiT** | Denoising Steps | 1000 | | | | |
| | Image Scaling | 0.8175 | | | | |
| | Patch Size | 2 | | | | |
| | Hidden Size | 768 | 384 | 768 | 768 | 768 |
| | Depth | 10 | 8 | 10 | 12 | 12 |
| | # Heads | 12 | 6 | 12 | 6 | 12 |
| | Learning Rate | 1e-4 | | | | |

Table 5: Comparison results of image generation.

| | Model | CLEVRTEX | OCT-B | MOVi-C | MOVi-E | FFHQ |
|---|---|---|---|---|---|---|
| IS | LSD | 4.87 | 3.50 | **4.05** | 3.73 | **4.11** |
| | SlotDiffusion | 4.47 | 2.80 | 3.98 | 3.34 | 3.86 |
| | CODiT | **7.58** | **3.88** | **4.05** | **3.96** | 3.76 |
| FID | LSD | 117.14 | 41.85 | **90.34** | **70.23** | 57.26 |
| | SlotDiffusion | **48.04** | **35.37** | 101.84 | 83.43 | **50.02** |
| | CODiT | 83.39 | 79.28 | 138.37 | 127.64 | 67.54 |

## E.1 IMAGE GENERATION

In this section, we measure the generation ability of the proposed method by the metrics of IS and FID. Following LSD and SlotDiffusion, we adopt the same sampling strategy for all methods. Firstly, we collect the object slots encoded from train datasets images to get object slots set $S_{\text{train}}$. Then, we apply the K-means algorithm on the set to get $K$ clusters. Finally, we select one object slot randomly from each cluster and put it into the decoder to generate the complete image. We sampled 1000 images and compared them with the complete training dataset as the standard setting. The IS results and samples can be seen in Table 5 and Figure 6 respectively.

The results show that CODiT achieves the highest IS among all datasets except FFHQ. As we use the composition decoder during generation, the objects in the scenes are less dependent on each other than the methods with pre-decoder compositional denoising network, which enables the former to generate more variant images. We also notice that if the number and size of objects are minor (such as in CLEVRTEX and OCT-B), both LSD and SlotDiffusion tend to duplicate these objects, which

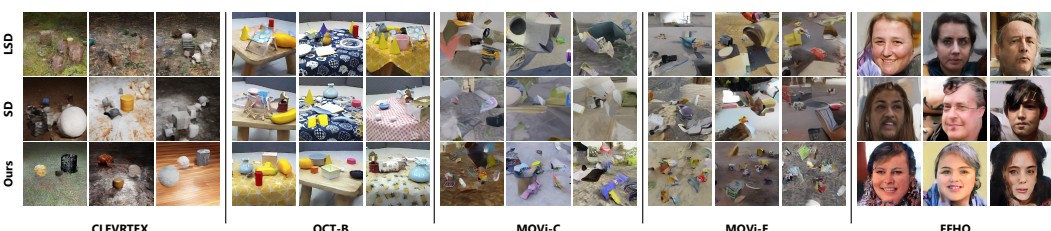

Figure 6: The sampled image of the proposed method, LSD, and SlotDiffusion.

Table 6: Comparison results of CODiT, BO-QSA, and their combination 'CODiT+BO-QSA', where we replace the Slot Attention encoder of CODiT with the encoder in BO-QSA.

| Dataset | Model | ARI-A↑ | ARI-O↑ | AMI-A↑ | AMI-O↑ | mIOU↑ | mBO↑ |
|---------|-------|--------|--------|--------|--------|-------|------|
| CLEVRTEX | BO-QSA | 12.50 | 55.56 | 28.74 | 58.11 | 32.36 | 35.95 |
| | CODiT | **77.48** | 90.81 | **70.44** | 91.54 | **62.37** | **67.82** |
| | CODiT+BO-QSA | 46.15 | **93.17** | 48.93 | **93.95** | 44.86 | 51.45 |
| OCT-B | BO-QSA | 17.61 | 55.36 | 44.06 | 61.21 | 45.27 | 45.32 |
| | CODiT | **74.96** | **79.29** | **68.40** | **82.50** | **51.63** | **56.60** |
| | CODiT+BO-QSA | 25.05 | 69.05 | 47.40 | 72.77 | 37.89 | 40.98 |

leads to abnormal parts of images, which will be discussed specifically in Section 4.1. For FFHQ, since the concept of objects is unclear and dependent on each other spatially, randomly choosing object slots to compose may cause spatial incongruity compared with non-compositional methods.

It is also worth noting that CODiT does not perform well in FID. We mainly attribute this to two reasons: First, as described in CFG (Ho & Salimans, 2022), better FID usually leads to worse IS. The result shown in Table 5 is also consistent with the original paper. Second, the compared methods (LSD, SlotDiffusion) mainly adopt FID to measure their generation ability. Compared with IS, FID mainly measures the similarity between the generated images and the training set. As the compared methods predict the added noise in the entire image latent directly, it is easier for them to capture the feature of the entire image, making the generated images more similar to the images from the training set, and finally obtain better FID. In comparison, CODiT is more likely to learn the individual object representations instead of the relation between them. As a result, CODiT generates images that are novel but less similar to the training set, leading to worse FID but better IS. Since we focus more on the compositionally of the methods as well as the independence of the object slots, we think that IS is more suitable for measuring the contribution of the proposed method.

### E.2 COMBINATION CODiT WITH BO-QSA

As introduced in Section 2, BO-QSA (Jia et al., 2022) introduces a bi-level optimization technique to the original Slot Attention encoder. Since these two models have a similar encoder architecture, We can replace the Slot Attention encoder in CODiT with the encoder in BO-QSA, aiming to improve the model performance. We also conduct experiments on the original BO-QSA model for comparison. Results in Table 6, show that the BO-QSA architecture does not bring obvious improvements to CODiT. We found that such a design causes the model to segment scenes into several parts if the image resolution is large(256×256), which can also be seen in the original paper. However, with the combination of CODiT and BO-QSA, the model still represents a great ability to segment foreground objects in CLEVRTEX, which is worth exploring in our future work.

### E.3 CONTROLLABLE COMPOSITIONAL GENERATION

To show the ability of compositional generation, following LSD, we use the representations from different images to generate the whole image. We test our model mainly in CLEVRTEX, OCT-

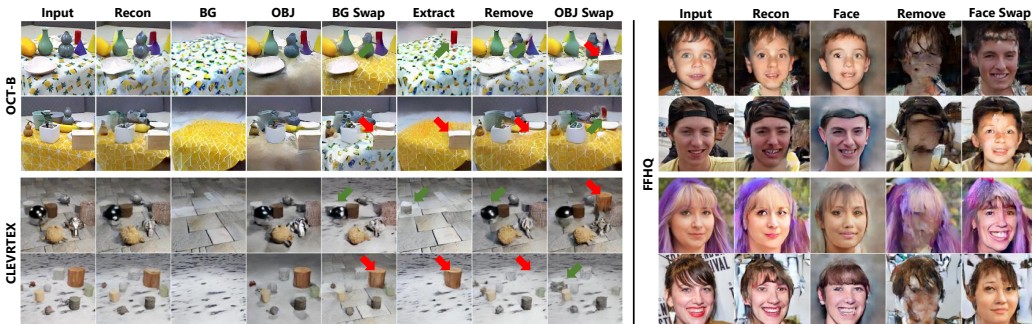

Figure 7: Controllable compositional generation results of CODiT on CLEVRTEX, OCT-B, and FFHQ. For CLEVRTEX and OCT-B, We show the generation results under different types of control including background extraction(BG), foreground extraction(OBJ), background exchange(BG Swap), single object extraction(Extract), single object removal(Remove), and object exchange(OBJ Swap). The exchanged objects are pointed by red or green arrows respectively. For FFHQ, we mainly choose two images and change their face areas to compose novel portraits.

B, and FFHQ in this section. For CLEVRTEX and OCT-B, we first choose two images from test datasets randomly and obtain objects or background representations. We treat the representation that has the largest generation mask as background and the others as foreground objects. During generation, we exchanged the background representations or random-selected object representations of two images, and we also tried other related manipulation ways including removing the background. For FFHQ, we mainly choose two images and change their face areas to compose novel portraits. The results in Figure 7 show that our object representations are reusable to generate different images.

### E.4 COMPUTATIONAL COMPLEXITY

Table 7: Comparison results of computational complexity during forward propagation for one step. All of the results are obtained in the CLEVRTEX dataset and on a single GeForce RTX 4090 GPU device, where we set the batch size to 1.

|  | Memory(Gflops) | Time(s) | Model Size(MB) |
|---|---|---|---|
| LSD | 942.08 | 0.045 | 113 |
| CODiT | 1730.04 | 0.052 | 175 |

We reported the computation complexity of CODiT and LSD during training in Table 7. Although the computational complexity of our method grows linearly with the number of slots, it should be pointed out that the decoder based on the cross-attention mechanism also grows linearly with the number of slots. Thanks to the parallel computing capability of the GPU, even if the number of our model parameters is higher than that of the comparative method, our model can still achieve similar computational efficiency in the decoding stage to LSD, while the latter has limited efficiency improvement through parallel computing.

### E.5 OBJECT PROPERTY PREDICTING

Following Slot Attention (Locatello et al., 2020) and LSD (Jiang et al., 2023), we also measure the ability of CODiT to predict object properties on CLEVRTEX. As shown in Table 8, CODiT can predict the object properties better than baselines. We think the result is reasonable, as shown in Figure 3, CODiT can learn object slots with better interpretability, therefore it is easier for it to predict these properties.

Table 8: Comparison results of the task of object property predicting.

| Properties | CODiT | LSD | SlotDiffusion |
|---|---|---|---|
| Material ↑ | **0.686** | 0.553 | 0.511 |
| Shape ↑ | **0.793** | 0.694 | 0.744 |
| Rotation ↓ | **0.209** | 0.510 | 0.652 |

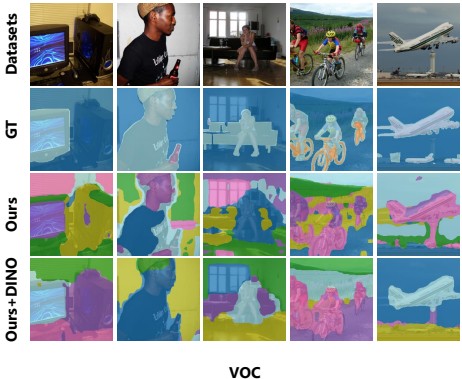 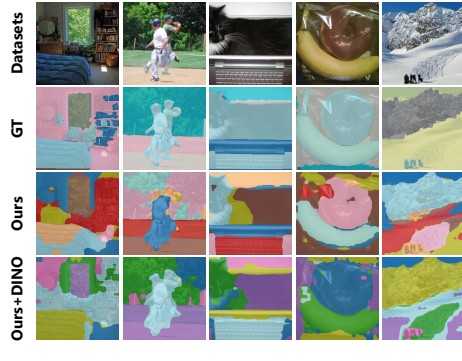

Figure 8: The segmentation results on VOC and COCO. It can be seen that although the plain CODiT does not perform well, with the equipment of CODiT, CODiT can achieve better segmentation results compared with DINOSAUR. Although some parts of the mask generated by CODiT equipped with DINO are inconsistent with the ground truth, they still represent meaningful targets.

### E.6 TO REAL WORLD COMPLEX DATASETS

Recent methods also aim to expand object-centric learning into real-world complex datasets. To show the scalability of CODiT, we also test the proposed method on VOC and COCO. To fully utilize the pre-trained DINO-ViT Encoder of DINOSAUR, which is designed for images with a resolution of 224×224, all images are resized and cropped to 224×224. The corresponding results can be seen in Table 9 and Figure 8, which shows that although the quantitative results of CODiT is not good, with the equipment of pre-trained DINO encoder, CODiT can achieve better segmentation result compared with the original one. We also notice that even with the pretained DINO, CODiT still not perform as well as these recent methods in terms of metrics. We mainly attribute it into two reasons: First, as the relation of objects in real-world datasets is more clear than synthetic datasets, the cross-attention mechanism in both autoregressive transformer decoder and UNet decoder makes it easier to capture this relation and can learn representations better. In comparison, CODiT composes the entire scene in a more independent way. Second, as also mentioned in (Wu et al., 2023), the objects in real-world data are not well-defined and faces severe part-whole ambiguity. It is also noticeable that CODiT equipped with a DINO encoder can discover areas that are not labeled but have clear semantic information, which proves the unrealizability of the official label in terms of unsupervised OCL methods.

## F FAILURE CASES

As described in Section 4.2, we can adopt an extra unconditional decoder to edit certain regions through CFG. However, we found that this ability may fail in face datasets FFHQ. As shown in Figure 9, if we edit certain regions of the whole image, some unnatural artifacts may appear. We attribute this to the connection between the slots in FFHQ being close, as a result, if we edit only one region independently, this may cause incoherency between these regions. Compared to CLEVR-TEX, FFHQ may not be appropriate for CODiT in terms of the object editing task since CODiT processes object slots more independently to extract representations with higher interpretability. A

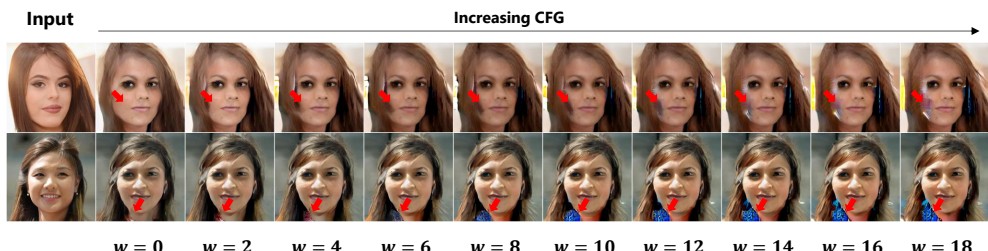

Figure 9: Failure cases of object editing experiments on FFHQ. We edit the hair region in the first row and the clothes region in the second row. Unnatural artifacts appear on the edge of faces as $w$ increases.

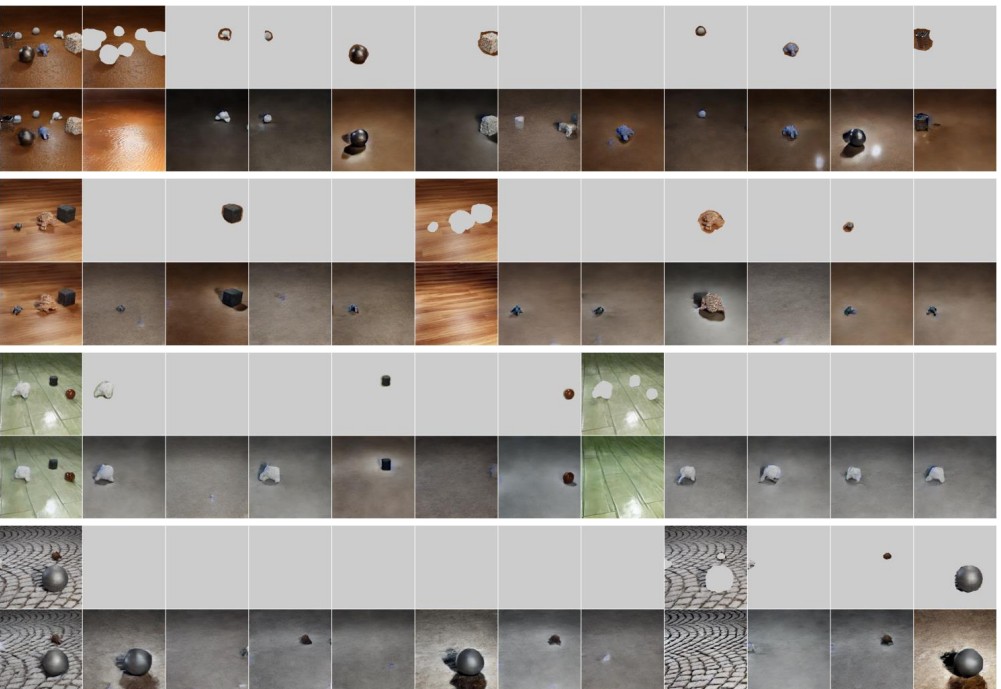

Figure 10: Individual object generation results of CODiT on CLEVRTEX.

possible method to solve this problem can be using more conservative CFG parameters, such as setting $w$ to 4 for Figure 9.

## G   MORE VISUALIZATION RESULTS OF EXPERIMENTS

We show more visualization results of experiments in this section. The corresponding results can be seen from Figure 10 to Figure 13.

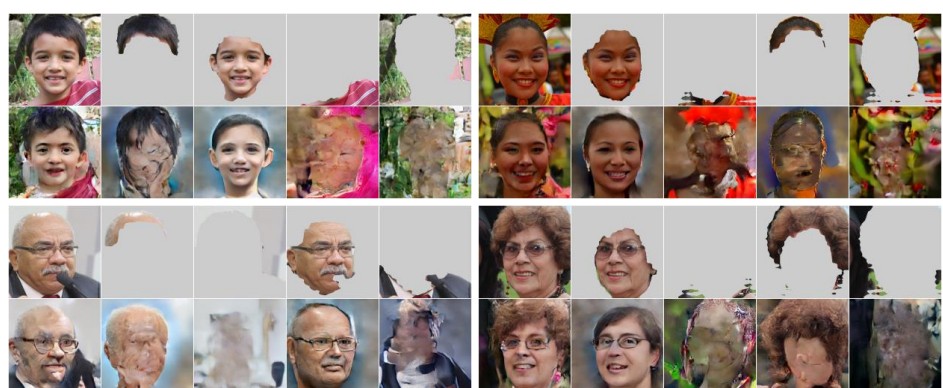

Figure 11: Individual object generation results of CODiT on FFHQ.

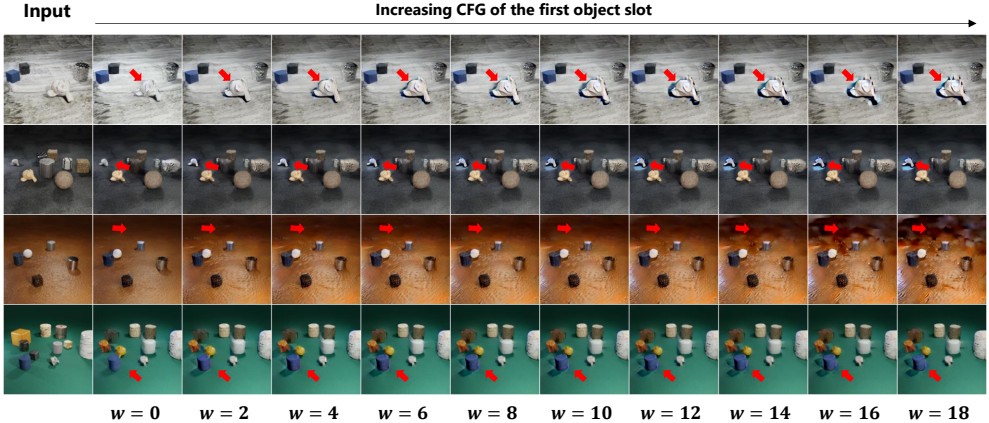

Figure 12: The visualization of object editing experiments.

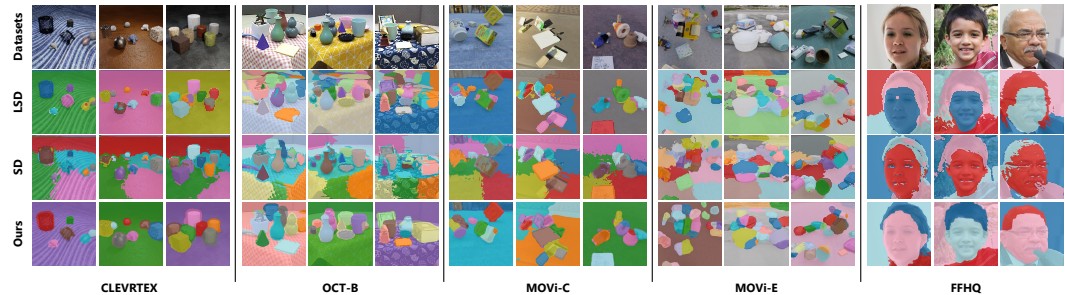

Figure 13: The segmentation results of LSD, SlotDiffusion, and CODiT.

Table 9: Results on real-world datasets. The results of DINOSAUR and DINOSAUR-MLP are copied from (Seitzer et al., 2022), the results of SPOT are copied from (Kakogeorgiou et al., 2024),the results of CAE are copied from (Löwe et al., 2022), the results of Rotating Features are copied from (Löwe et al., 2024), the results of SlotDiffusion+DINO ViT are copied from (Wu et al., 2023). 'CODiT+DINO ViT' represents the model where we replace the UNet backbone in the Slot Attention encoder of CODiT with a pre-trained DINO ViT which is frozen during training.

| | VOC | | COCO | |
|---|---|---|---|---|
| **Model** | **mBO$^i$** ↑ | **mBO$^c$** ↑ | **mBO$^i$** ↑ | **mBO$^c$** ↑ |
| DINOSAUR | 43.6 | 50.8 | 32.3 | 38.8 |
| DINOSAUR-MLP | 39.5 | 40.9 | 27.7 | 30.9 |
| SPOT | 48.3 | 55.6 | 35.0 | 44.7 |
| CAE | 32.9 | 37.4 | - | - |
| Rotating Features | 40.7 | 46.0 | - | - |
| SlotDiffusion+DINO ViT | 50.4 | 55.3 | 31.0 | 35.0 |
| CODiT | 35.2 | 39.0 | 22.4 | 27.4 |
| CODiT+DINO ViT | 45.8 | 51.6 | 28.1 | 34.1 |

