# OpenReview forum: "Compositional Scene Modeling with An Object-Centric Diffusion Transformer"
_ICLR.cc/2025/Conference — Submitted to ICLR 2025_

### Official Review · Reviewer_DL1z · 2024-10-19

**Soundness:** 2
**Presentation:** 3
**Contribution:** 2
**Rating:** 5
**Confidence:** 5

**Summary:**

This paper proposes CODiT, a DiT-based model for object-centric learning (OCL). CODiT adopts a post-decoder composition design, i.e., it runs a shared-weight DiT on each slot to predict a noise map and a mask in parallel. This is very different from recent works which use the pre-decoder composition (e.g., LSD, SlotDiffusion). In addition, the paper points out that such design is related to the classifier-free guidance (CFG) in diffusion model sampling. The authors conduct experiments on three synthetic and four real-world datasets to evaluate CODiT.

**Strengths:**

- As far as I know, this paper is the first to study individual object generation ability of OCL models. It surprising to see that prior works perform so badly in this aspect. It serves as a good motivation for this paper.
- The analogy to CFG is a new perspective to OCL models, especially the fact that object masks enable different CFG values at different pixels. Indeed, this is very intuitive and reasonable in the context of object-centric representations.

**Weaknesses:**

- For the image reconstruction result in Sec. 4.4, why only reporting MSE? I think prior works also report LPIPS or FID, which are more aligned with human perception.
- Another important aspect of OCL methods is the learned slot representations. While I would expect the representation quality to be good (given the single-object generation result), the paper should still show some results in this direction. For example, the object property prediction task in LSD, or the downstream VQA task in SlotDiffusion.
- For the real-world image segmentation result in Appendix E, why is SlotDiffusion not included here? Its performance is clearly higher than CODiT + DINO. Also, please include COCO as it is also a standard benchmark in prior works.
- The related work section is not comprehensive enough. OCL is a broad field, but this paper only has ~35 citations. Please refer to the related work section of LSD or SlotDiffusion to add more discussion about prior works.

**Questions:**

- The compositional generation results in Fig.7/8/9 are quite blurry compared to Fig.6/10. Can authors provide a clearer version? This is important as compositionality is a crucial aspect of OCL methods.
- Line 365 says "we use image-based SlotDiffusion for fair comparison". What is the "image-based" version? SlotDiffusion always use a VQ-VAE and perform denoising in the latent space.

---

### Official Review · Reviewer_pxiR · 2024-10-30

**Soundness:** 2
**Presentation:** 2
**Contribution:** 2
**Rating:** 6
**Confidence:** 4

**Summary:**

This paper introduces a new method to compose slot components which is different from common methods used in previous object-centric learning approaches. Traditionally, object-centric learning either uses separate decoders for individual slot components followed by mask-sum operation or a single diffusion decoder taking all slot components as conditionings to reconstruct input images. This paper combines the both and proposes to use separate diffusion decoders with shared parameters for individual slot components followed by mask-sum operation in the score (noise prediction) domain. The paper claims that their approach, CODiT, works better than traditional mask-sum approaches for complex scenes and outperforms traditional diffusion decoder approaches in terms of slot representation interpretability and segmentation performance.

**Strengths:**

1. The paper points out an interesting finding from state-of-the-art approaches LSD [1] and SlotDiffusion [2], i.e., the decoded component of a single slot doesn't essentially correspond to a semantic local part (object or facial feature) of input image. As a result, the slot representations learned with these approaches lack interpretability for human understanding.

2. The paper proposes a straightforward approach that can solve this problem while still allows preserving the powerful decoding ability of diffusion models.

3. The paper provides extensive experiments to support the claims and show better segmentation and interpretability performance.

References

[1] Jiang, "Object-Centric Slot Diffusion", NeurIPS 2023

[2] Wu,  "Slotdiffusion: Object-centric generative modeling with diffusion models", NeurIPS 2023

**Weaknesses:**

1. Mask-sum operation indeed can provide better spatial segmentation for interpretability, but on the other hand makes the image rendering process less flexible. Since the masks define clear boundaries, when you stitch different components (either pixel or score) together with masks there could be unnatural artifacts along boundaries leading to blurry results. As observed in Fig 7 in the paper, when you compose different slot components either from same scene (reconstruction) or from different scenes (editing), the obtained results are far from natural and high-fidelity. In contrast, if you observe the editing and generation without using mask-sum operation in LSD Fig 3 Fig 4, and Fig 10 https://arxiv.org/pdf/2303.10834, they have much better generation performance. In summary, by introducing mask-sum operation, the approach actually trades generation performance for interpretability. It would be fair to explicitly point out this in the limitation section.

2. By summing predicted scores (noise predictions), the approach is actually closely related to Decomp Diffusion [2]. The only two main differences are: (1) CODiT uses slot attention for slot representation learning while Decomp Diffusion can use CNN or slot attention or any encoders. (2)  CODiT uses masks to sum noise predictions while Decomp Diffusion sum noise predictions without masks-sum operation. Again, adopting mask-sum operation limits the generation ability of CODiT when compared to Decomp Diffusion. Such an interpretability-segmentation trade-off should be addressed or pointed out in the paper as said above.

3. Image reconstruction evaluation. It is a bit confused that considering that the Fig 7 in the paper doesn't show strong image reconstruction performance while LSD or SlotDiffusion can actually reconstruct input images very well in their original paper, then why CODiT could have better MSE performance than LSD or SlotDiffusion? Did you also examine FID for reconstructions or edited images like what are done in LSD or SlotDiffusion?

4. Segmentation evaluation. This paper uses metric ARI-A and ARI-O among others for segmentation evaluation and states ARI-O considers foreground pixels only. In that case, is ARI-O same as FG-ARI commonly used in existing works? If so, why the ARI-O of baseline LSD is largely different from that in the original LSD paper? If ARI-O and FG-ARI are different, is there any reason why FG-ARI is not involved considering it is a de facto metric in this research topic?

5. The paper overclaims the image editing ability of the proposed approach. Actually, CODiT only outperforms LSD or SlotDiffusion in terms of individual component guidance, while the cross-image editing (swap, removal) ability shown in Fig 7 in the paper is even not comparable with LSD. Furthermore, for compositional generation with individual component controllability, GNM [4], Slot-VAE [5] and NLoTM [6] actually deomstrate such an ability and even with more flexibility where color shape or position can be controlled in a disentangle way. Though not directly comparable, discussing the difference of CODiT with [4][5][6] is helpful to put the approach into context.

References

[1] Jiang, "Object-Centric Slot Diffusion", NeurIPS 2023

[2] Su, "Compositional Image Decomposition with Diffusion Models", ICML 2024

[3] Wu,  "Slotdiffusion: Object-centric generative modeling with diffusion models", NeurIPS 2023

[4] Jiang, "Generative Neurosymbolic Machines", NeurIPS 2020

[5] Wang, "Slot-VAE: Object-Centric Scene Generation with Slot Attention", ICML 2023

[6] Wu, "NEURAL LANGUAGE OF THOUGHT MODELS", ICLR 2024

**Questions:**

Fig 6 actually shows some interesting results. Do you train a diffusion decoder or use a pretrained diffusion model like in LSD?

---

### Official Review · Reviewer_qGNY · 2024-11-03

**Soundness:** 3
**Presentation:** 2
**Contribution:** 2
**Rating:** 3
**Confidence:** 3

**Summary:**

The paper introduces CODiT, an object-centric learning framework that incorporates a post-decoder compositional diffusion network to improve the interpretability and generation capabilities of scene modeling. CODiT leverages a compositional denoising approach where individual object representations are denoised separately and integrated compositionally. This is differences from the existing pre-decoder methods that struggle with interpretability. The method demonstrates favorable performance in segmentation, reconstruction, and object-editing tasks.

**Strengths:**

-	CODiT’s integration of compositional modeling within a diffusion framework show a meaningful improvement in object-centric learning.

-	Human Intuition Alignment: The post-decoder compositional design aligns more closely with human visual scene processing.

-	The model’s performance on evaluation datasets (CLEVRTEX, OCT-B, and FFHQ) validate the methods applicability in both synthetic and real data.

**Weaknesses:**

### Incomplete literature survey

Although there have been recent advances in compositional image analysis using diffusion models, the paper does not adequately introduce or contrast its contributions with these newer works.

[1] Kakogeorgiou, Ioannis, et al. "SPOT: Self-Training with Patch-Order Permutation for Object-Centric Learning with Autoregressive Transformers." Proceedings of the IEEE/CVF Conference on Computer Vision and Pattern Recognition. 2024.

[2] Zadaianchuk, Andrii, Maximilian Seitzer, and Georg Martius. "Object-centric learning for real-world videos by predicting temporal feature similarities." Advances in Neural Information Processing Systems 36 (2024).


### Incomplete comparison with recent methods

The paper does not include a comparison or discussion involving recent relevant methods, such as [1]


### Limited evaluation benchmarks

While segmentation metrics are robust, it would be helpful to have a more extensive evaluation of real-world datasets such as PASCAL VOC and MS COCO.

### Limited Analysis of Failure Cases

The paper does not include failure cases or comprehensive visualizations where CODiT underperforms.

**Questions:**

### Comment

The main contributions that set CODiT apart from previous methods are not emphasized enough. Including a dedicated paragraph that explicitly contrasts CODiT with prior works would enhance readability and clarify its unique contributions. Currently, the advantages of CODiT are scattered across different sections, making it challenging to discern what specifically differentiates it from recent approaches.

---

> ### Comment · Area_Chair_AeHS · 2024-11-26
> **[ACTION NEEDED] Respond to author rebuttal**
>
> Dear Reviewer,
>
> Now that the authors have posted their rebuttal, please take a moment and check whether your concerns were addressed. At your earliest convenience, please post a response and update your review, at a minimum acknowledging that you have read your rebuttal.
>
> Thank you,
> --Your AC

---

### Official Review · Reviewer_Aa4N · 2024-11-05

**Soundness:** 3
**Presentation:** 2
**Contribution:** 2
**Rating:** 5
**Confidence:** 3

**Summary:**

This paper presents a compositional approach for learning interpretable object representations, which can then be used for object editing in images or for unsupervised segmentation. The method is similar to SlotDiffusion, but CODiT explicitly models object masks during the diffusion denoising stage. Results show improvements compared to SlotDiffusion and LSD on object editing and unsupervised segmentation tasks, as well as compared to DINOSAUR (another recent method) on segmentation tasks.

**Strengths:**

* I appreciate that the authors shared the model code in the supplementary (though I haven’t had a thorough look at it)
* Figure 1 nicely lays out the difference between this approach and many prior approaches
* Results look good compared to the baselines presented in the paper

**Weaknesses:**

1. A key weakness is I found a baseline after a cursory search that should have been included in related work and in the comparisons: Kakogeorgiou et al., CVPR 2024 (“SPOT: Self-Training with Patch-Order Permutation for Object-Centric Learning with Autoregressive Transformers”). This task is not my area of expertise, but this paper also evaluates on the same tasks as the current submission. On VOC unsupervised segmentation, it appears to outperform the proposed method (Table 5 in submission vs. Table 5 in Kakogeorgiou et al). I would appreciate a comparison of this work to that, both in method and in the experimental results.
2. I find that the intro and writing focus heavily on human intuition, and could do a better job of describing the task at hand and the relevance of huma intuition to accomplishing that task. Until the experiments section, it wasn’t clear to me whether the goal was to learn object representations (if so, for what? Classification? Segmentation?), segment objects, generate new images, or edit existing images. I found that the intro from prior work helped me understand the problem setting better, e.g. Seitzer et al., 2023 (DINOSAUR).
3. The method figures are generally understandable, but could use a polishing pass for aesthetics. Again, I’d refer to prior work like Seitzer et al, 2023, or Kakogeorgiou et al., 2024.
4. The figures for qualitative methods are quite hard to parse, and low resolution even after zooming in. I’d encourage the authors to increase the resolution here and the size of the figures.

**Questions:**

1. I see prior methods have also reported results on COCO, which is likely to be more challenging than PASCAL. Have you considered reporting results on COCO, or would it be possible to report results on it?
2. I’d request the authors to address my concerns in the weaknesses.
3. In particular, I’d like to see a comparison to Kakogeorgious et al.

---

> ### Comment · Area_Chair_AeHS · 2024-11-26
> **[ACTION NEEDED] Respond to author rebuttal**
>
> Dear Reviewer,
>
> Now that the authors have posted their rebuttal, please take a moment and check whether your concerns were addressed. At your earliest convenience, please post a response and update your review, at a minimum acknowledging that you have read the rebuttal.
>
> Thank you,
> --Your AC

---

### Meta-Review · Area_Chair_AeHS · 2024-12-19

**Metareview:**

This paper proposes a novel decoder for object-centric models using a compositional diffusion approach. While the approach is novel, the reviewer consensus was that the paper is not ready for publication. The authors are encouraged to take the reviewer feedback into account should they prepare a resubmission at a future venue.

**Additional Comments On Reviewer Discussion:**

No reviewer was willing to champion the paper for acceptance.

---

### Decision · Program_Chairs · 2025-01-22

Reject